# Quantitative assessment can stabilize indirect reciprocity under imperfect information

Laura Schmid [1] ✉, Farbod Ekbatani[2], Christian Hilbe [3] & Krishnendu Chatterjee[4]

The field of indirect reciprocity investigates how social norms can foster cooperation when individuals continuously monitor and assess each other's social interactions. By adhering to certain social norms, cooperating individuals can improve their reputation and, in turn, receive benefits from others. Eight social norms, known as the "leading eight," have been shown to effectively promote the evolution of cooperation as long as information is public and reliable. These norms categorize group members as either 'good' or 'bad'. In this study, we examine a scenario where individuals instead assign nuanced reputation scores to each other, and only cooperate with those whose reputation exceeds a certain threshold. We find both analytically and through simulations that such quantitative assessments are error-correcting, thus facilitating cooperation in situations where information is private and unreliable. Moreover, our results identify four specific norms that are robust to such conditions, and may be relevant for helping to sustain cooperation in natural populations.

Reputation-based social dynamics are a fundamental feature of the human experience[1–5]. When social interactions are observed and judged by other community members, individuals start to treat their reputation as a valuable commodity[6,7]. They are more inclined to help others because this increases their chance to receive such help themselves[8–10]. In evolutionary game theory, this mechanism for cooperation is referred to as "indirect reciprocity".[11–15]. In contrast to direct reciprocity[16], this mechanism does not require repeated interactions between the same two individuals. Instead, people learn to cooperate to increase their public standing, which can be valuable in future interactions with third parties. Indirect reciprocity and its impact on human behavior hence tie into a wide range of fields, such as social psychology[17–19], moral philosophy[1,20], or online reputation systems, marketing, and public relations[21–23].

To model the effect of indirect reciprocity, researchers consider a particularly simple social interaction, the donation game[13–15]. In this game, a donor chooses whether or not to pay a small cost to confer a benefit to someone else, i.e., whether to cooperate or to defect. Other population members observe the donor's action and they update the donor's reputation accordingly. How this updating occurs depends on a community's social norm. Norms consist of two components, an assessment rule and an action rule[24]. Assessment rules prescribe how to update reputations based on observations. They specify which behaviors should improve a donor's reputation, and which behaviors should be condemned. Action rules tell the donor whether or not to cooperate with a given recipient.

The literature on indirect reciprocity explores which social norms can sustain cooperation, and how complex such norms need to be[25–30]. Early work stresses the effectiveness of simple norms, such as 'Image Scoring'[25,26]. When communities use this norm, any cooperative action leads to an increased reputation, whereas any defection reduces the donor's score. According to the respective action rule, donors should only cooperate with those recipients whose score is above a certain threshold. Image scoring is an instance of a so-called "first-order norm". Here, assessments only depend on the donor's action. Such first-order norms, however, have been suggested to be unstable[29–32].

[1]KAIST Graduate School of AI, 02455 Seoul, South Korea. [2]Booth School of Business, The University of Chicago, Chicago, IL 60637, USA. [3]Max Planck Research Group Dynamics of Social Behavior, Max Planck Institute for Evolutionary Biology, 24306 Plön, Germany. [4]IST Austria, Am Campus 1, 3400 Klosterneuburg, Austria. ✉ e-mail: laura.schmid@kaist.ac.kr

After all, individuals have no incentive to withhold help from a recipient with a low score, since doing so would harm their own reputation. As a way to resolve this weakness, the literature argues that social norms should differentiate between justified and unjustified defections. To do so, the respective assessment rules need to take into account the reputation of the recipient ("second-order norms")[6,30,31], or even the previous reputation of the donor ("third-order norms")[27,33]. Such norms might suggest that although defecting against a well-reputed player is bad, the same behavior against an ill-reputed player is acceptable.

In order to systematically compare the performance of different norms, Ohtsuki and Iwasa implemented an exhaustive search among all binary social norms, such that individuals are either "good" or "bad". In their landmark papers[28,33], they describe all social norms up to the third-order that can stabilize cooperation. To this end, they consider reputations to be public and synchronized. This assumption implies that all members of the population update a donor's assessment in the same way; two different players never disagree on the assessment of a third party's reputation. Such synchronized reputations naturally arise when all assessments are made by a central authority[34]. Using this setup, Ohtsuki and Iwasa identify eight particularly successful social norms, termed the "leading eight". If employed by the entire population, each of these eight norms can maintain full cooperation and resist the invasion of free riders. The assessment and action rules of these eight norms exhibit several interesting patterns (Fig. 1a). For example, a good donor who cooperates with a good recipient should always obtain a good reputation. Moreover, any donor who defects against a good recipient should end up with a bad reputation. The different leading eight norms disagree, however, on how to assess actions towards bad recipients. For example, while some norms suggest that any form of cooperation should yield a good reputation, others like "Stern Judging" disincentivize cooperating with an ill-reputed recipient.

The premise of public and synchronized information facilitates a rigorous analysis of social norms[35–40]. At the same time, however, the premise appears to be a strong idealization: individuals do not always agree on others' reputations, nor do they usually have access to the same information. Previous work has shown that once players make their own private judgments, the leading eight may no longer sustain cooperation[41–43]. In that case, disagreements can arise because some individuals may not observe certain interactions, or they may misinterpret an interaction's outcome[44]. Such disagreements can proliferate, leading to separate sub-communities who consider each other as bad, even though everyone employs the same social norm[45]. To resolve this sensitivity of the leading eight with respect to private and noisy information, previous approaches have ranged from finding new and potentially simpler strategies of indirect reciprocity[46–50], to adding elements of empathy or "pleasing" behaviors to higher-order norms[51,52].

In contrast, here we show that the leading eight norms naturally become more robust as a consequence of modeling reputations to be more fine-grained. By introducing quantitative assessment into higher-order strategies, we incorporate an important ingredient of many natural reputation systems, namely that reputations come in different degrees of goodness. That is, individuals' opinions about each other can take values beyond "good" or "bad". This modeling choice is not only a useful premise in many real-life applications. Previous theoretical research has suggested that more refined reputation scores can act as a buffer for errors by increasing a social norm's tolerance with respect to a small number of bad actions[48,49]. Furthermore, replacing binary reputations with continuous variables has been shown to facilitate cooperation when also the player's feasible actions are continuous[53].

In our model, individuals keep track of each others' reputations in the form of integer scores, taking values from a fixed range. For an

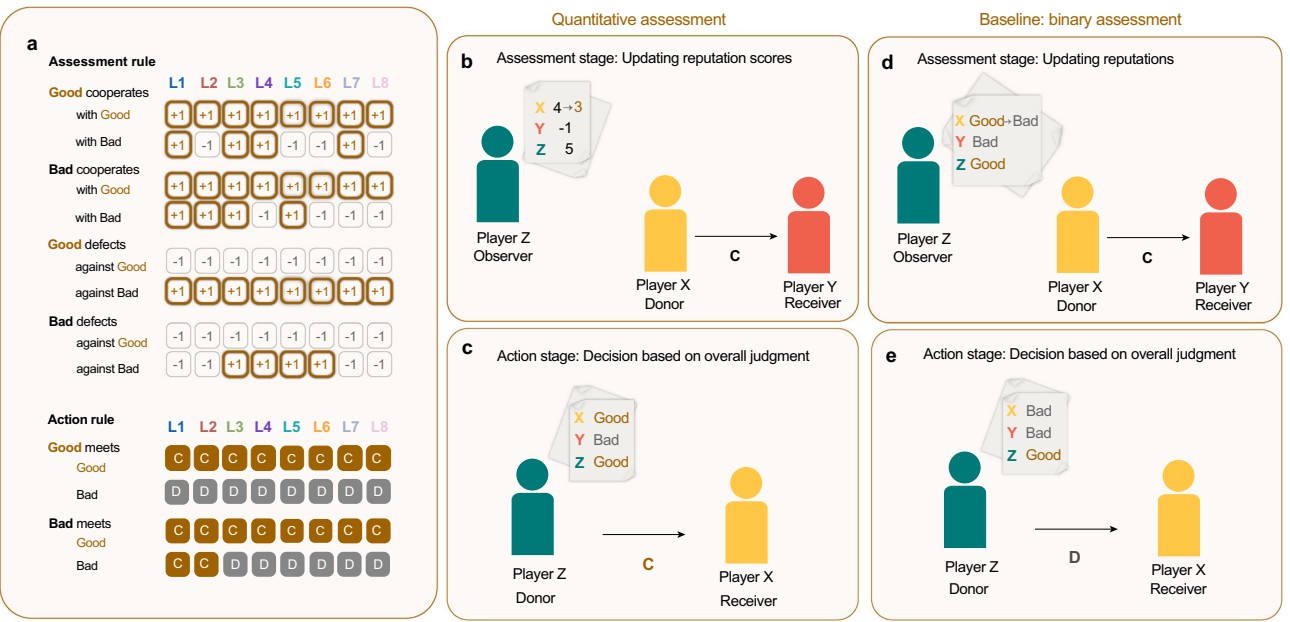

Fig. 1 | The leading eight norms with quantitative assessment. a We consider the leading eight norms[28,33]. Each norm consists of an assessment rule that determines how an observer updates a donor's reputation, and an action rule that governs players' behavior when they are chosen to be the donor. The assessment rule takes the context of an observed action into account: how an observer judges a donor depends on the donors' and recipient's reputation. Similarly, the action rule uses the current donor's and recipient's reputation to decide whether the donor should cooperate with the recipient. In contrast to the original baseline model, we now interpret a positive assessment of an action as an increment of +1 to the donor's

reputation score, and a negative assessment as a decrement of -1. b The observing Player Z assesses Player X's action of cooperating with a bad player as bad, such that he decrements X's previous score by one. c When it is Player Z's turn to be the donor, he translates his and Player X's reputation score into a binary label of "good" or "bad". Since both his and Player X's score are above the threshold S, he judges both himself and Player X as good, and cooperates. d, e In the baseline model using binary assessment, the same starting scenario ends differently: Player Z changes his view of Player X from good to bad after Player X cooperates with Player Y, and therefore defects against Player X.

observer, positively assessing a donor's action then means to increment the donor's score, while a negative assessment translates into decrementing the score (Fig. 1b). When deciding whether to cooperate with a given co-player, individuals translate these integer scores into the binary variables of "good" and "bad" (Fig. 1c). Individuals with a score above a pre-determined threshold are considered as good, whereas all others are considered bad.

Our agent-based simulations show that quantitative assessments effectively reduce the number of disagreements between population members. Consequently, individuals are better able to single out defectors while maintaining a good self-image. In fact, we observe that by shifting from binary thinking to more nuanced reputations amongst players, four of the leading eight norms become particularly robust: In contrast to the traditional setting with binary assessments, we find that each of these four rules now achieves high cooperation rates well above 75% even at significant noise levels of 10%. Our additional mathematical analysis further substantiates these simulation results. We show that homogeneous populations of leading eight players with quantitative assessment can efficiently recover from disagreements. In this way, we further highlight the strong error-correcting properties of more nuanced reputations. Finally, we formally characterize necessary properties for the evolutionary success of a social norm under imperfect information. This characterization leads us to identify the four norms previously found to be successful in our evolutionary simulations.

Overall, our results demonstrate the importance of nuanced assessments when interacting in communities with complex social norms. In a noisy environment with scarce information, binary judgments can irrevocably harm cooperation. Instead, reputation systems work best when reputations are sufficiently fine-grained.

## Results

**Game dynamics.** Similar to previous work, we consider a well-mixed population of fixed-size $N$. The members of this population, who we refer to as players, are continuously engaged in pairwise interactions. In every round of this series, two players, a donor and a recipient, are drawn at random from this population for an interaction. The donor chooses whether or not to confer a benefit $b$ to the recipient at his own cost $c < b$, i.e., chooses between cooperation and defection. The donor's choice is independently observed by other members of the population with probability $q$. They use these observations to privately update their opinion of the donor, without publicly sharing their judgment. Every player individually tracks the reputations of all other members of the population. We furthermore assume that players' observations are subject to noise: with probability $\varepsilon$, an observer mistakes a defection for cooperation, or a cooperative action for a defection. Given that information is thus assumed to be private and noisy, it is important to note that when two observers differ in their initial assessment of a given donor, they may also disagree on the donor's updated reputation, even if both observe the same interaction.

Every player is equipped with a social norm (corresponding to their "strategy"). Social norms govern the players' behaviors, both in terms how they act against one another as well as how they judge each other's actions. To fulfill this function, social norms consist of two components, an assessment and an action rule[15]. The assessment rule prescribes how to update a donor's reputation after observing their action and taking this action's context into account. The action rule prescribes whether to cooperate or defect in a given situation.

In line with Ohtsuki and Iwasa's original work[28,33], we consider norms of at most third-order. This means that assessments can depend on the donor's decision whether to cooperate, and both the donor's and the recipient's previous reputation. For deciding whether to cooperate, the donor's own reputation and the recipient's reputation are the determining factors. In our framework, reputations are not binary; instead players assign integer reputation scores $r$ to each other

that can take a wider range of values on a scale from a lower limit $V$ to an upper limit $A$. That is, we equip higher-order norms with quantitative assessment (Fig. 1a). While this leaves the considered leading eight norms $L1 - L8$ formally unchanged, the interpretation of the rules making up the norm and therefore the resulting reputation dynamics differ from the baseline model[44].

We note that there are two components to our model of quantitative assessment. For one, the integer reputation scores are used by the players to track changes in opinion about others in a fine-grained manner. We assume that if an action is judged as good based on an observer's assessment rule, the donor's score in the eyes of that observer is increased by one, whereas if the action is judged as bad, the donor's score is decreased by one (Fig. 1b). On the other hand, players need to be able to translate these nuanced scores into an overall judgment of a player in order to use the leading eight norms as defined in previous work. This overall judgment must result in a binary label to become the input bits for players' assessment and in particular action rules (Fig. 1c). To make the transformation unambiguous, we assume that players compare the reputation score of a given player with a threshold $S$ that separates scores into good and bad, similar to previous work on non-binary Image Scoring norms. An individual with a score equal to or above the threshold is judged as "good" overall, whereas a score below the threshold leads to an individual being judged as overall "bad". This means that the more refined the reputation scores, the less sensitive the overall label to single events, since this implies more shades of "good" and "bad". It is these overall reputation labels that are then used by players in order to correctly assess an observed action and to decide whether to cooperate with a specific recipient. We note that when we set the minimum score to $V = 0$, the maximum score to $A = 1$, and the cooperation threshold to $S = 1$, we recover the original baseline model of the leading eight with binary assessment (Fig. 1d, e).

In the following, we assume that all players use the same frame of reference for their assessment, such that everyone agrees on the assessment scale, i.e., the range of reputation scores. Each individual then uses the same fixed values for $V$, $A$, and $S$ throughout. No player has a more nuanced view of their environment than another, and no player judges others' scores with a different measure than another. Furthermore, we assume that the possible range of reputation scores is symmetric, with $A = R$, $V = -R$, and a threshold of $S = 0$. We denote a particular setting of players' frame of reference by $R$.

**Analyzing reputation dynamics.** We start testing our framework by first considering players' norms as fixed, if possibly different. Individuals play the donation game over many rounds and update others' reputation scores after every interaction. We can describe this change in players' reputation scores with time-dependent image matrices[41] $M(t) = r_{ij}(t)$, with $r \in [-R, R]$, representing the collection of players' reputation score repositories. These matrices thus record at any point in time which scores players assign to each other (Fig. 2a, left side). In every round of the game, $M(t)$ is updated according to the assessments of those players that have observed the donor's action, with their observation subject to noise $\varepsilon$. Noise in the environment implies that every observer has an independent probability $\varepsilon$ to misperceive the donor's action. If player $i$ assesses his observation of player $j$'s action as good, $r_{ij}(t+1) = r_{ij}(t) + 1$, otherwise, $r_{ij}(t+1) = r_{ij}(t) - 1$. In case the reputation score is already at the maximum $R$ (minimum $-R$), it simply keeps its value when the action is assessed as good (bad). To make the overall judgment of player $j$'s and $k$'s reputation, $i$ compares their reputation scores with the threshold $S$, and labels them "good" when $r_{ij} \geq S$ and "bad" when $r_{ij} < S$. This acts like a second, less refined layer of the reputation dynamics (Fig. 2a, right side). For example, if player $i$ judges donor $j$ to be good and recipient $k$ to be good, they will assess a defection of $j$ against $k$ as bad and decrease $j$'s score by one.

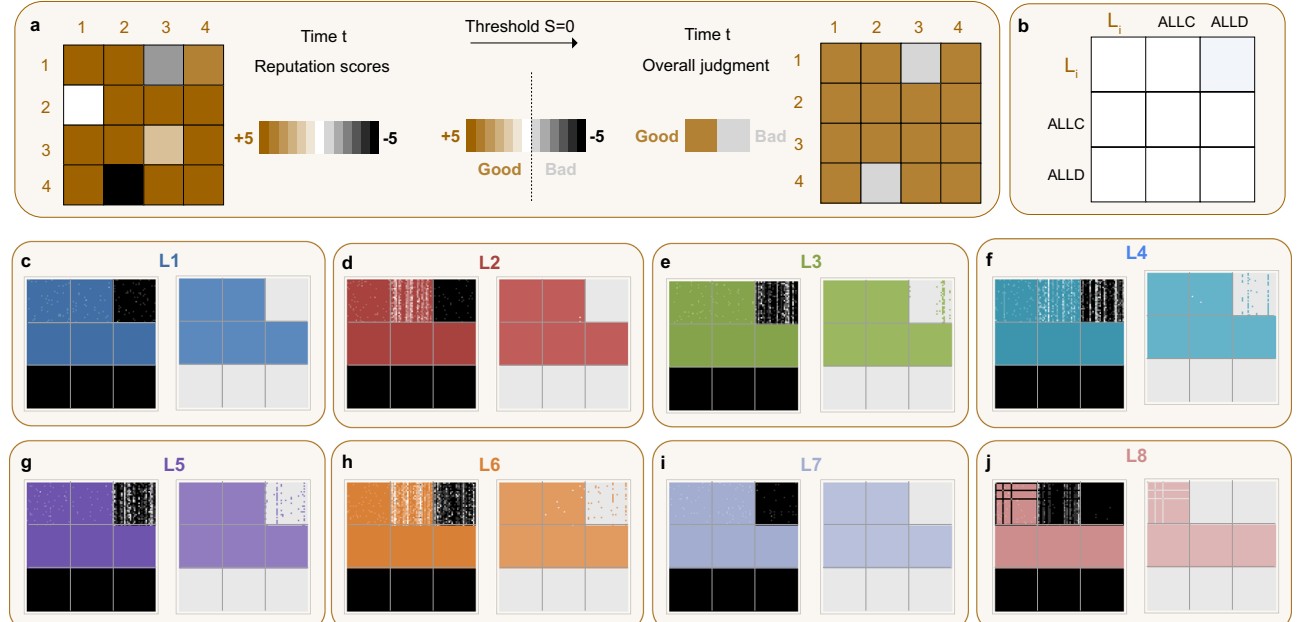

**Fig. 2 | Quantitative assessment and reputation dynamics. a** Image matrices represent how players assess each other at any given time. We assume that every player keeps track of each population member's reputation score, with scores in the interval [−5, +5]. To depict these image matrices graphically, we use colored dots, with the intensity of the color corresponding to the score: for example, a white dot means that the corresponding row player attributes a score of $r = 0$ to the corresponding column player (left side). On the other hand, players also make an overall judgment of others, in order to be able to use their assessment and action rules. To do so, they compare the scores to a threshold $S = 0$, resulting in a binary labeling of "good" and "bad". To visualize this second, less refined layer of the reputation dynamics, we use a matrix with colored and gray dots (right side). **b** We show image matrices when players either use a leading eight social norm $L_i$, ALLC, or ALLD (in equal proportions). **c–j** We show the snapshots at $T = 2 \times 10^6$ of players' reputation scores and binary labels they translate into for all leading eight norms. We see that for $L1$ (**c**) and $L7$ (**i**), the reputation assignments of different $L_i$ players are perfectly correlated. They assign only good reputations to each other and ALLC, while they only assign bad reputations to ALLD. The picture is very similar for $L2$ (**d**). For all other norms, there are disagreements among the $L_i$ players, where they can also perceive ALLD players favorably. We note that $L8$ does not perceive any ALLC player as good, which is one of two very stable states in the reputation dynamics. Parameters: Population size $N = 90$, error rate $\varepsilon = 0.05$, observation probability $q = 0.9$, frame of reference $R = 5$ (i.e., interval for reputation scores $r \in [−5, 5]$). Threshold $S = 0$. Simulations are run for $2 \times 10^6$ iterations, and the initial image matrix assumes a good reputation for all players.

We illustrate this concept with an example. We consider a population consisting in equal proportions of players using one of three social norms, a leading eight norm $L_i$, ALLD and ALLC (Fig. 2b). ALLD is the norm that prescribes to assign a bad reputation to everyone else and unconditionally defect, whereas ALLC prescribes to assign a good reputation to everyone else and unconditionally cooperate. For this population, we explore the reputation dynamics when information is noisy and scarce. In Fig. 2c–j, we present two snapshots of image matrices at the same timestep for each of the norms $L_1 - L_8$, where players use an assessment scale from −5 to 5. The first snapshot visualizes the integer reputation scores that players assign to each other at this particular time. The other snapshot represents the overall judgment that emerges when players compare others' scores with the given threshold ($S = 0$). The resulting binary label is then an indicator of how more refined assessment can help protect reputations from the negative effects of disagreements. We visualize the difference between the impact of the baseline binary assessment and quantitative assessment on reputation dynamics in Fig. 3. In this figure, we present the average images the three norms $L_i$, ALLC, and ALLD have of each other, for each of the $L_i$. We find that, compared to the baseline model, quantitative assessment helps all leading eight norms assign more accurate reputations to population members. It clearly improves not only the self-image of each leading eight norm, but also enhances their ability to distinguish between the two other strategies they compete with. In particular, singling out defectors becomes much easier to do. We find that $L1$ (Fig. 3a) and $L7$ (Fig. 3g) excel most at assigning appropriate reputations to their co-players despite the presence of noise. They are most likely to assign a good reputation to other players of their own kind and to ALLC, and

to assign a bad reputation to ALLD. The norm $L2$ (Fig. 3b) also does very well, even if it sometimes judges an ALLC player as bad (which however also happens without noise). Even $L3 - L6$ (Fig. 3c–f), which have particular trouble to accurately label defectors as bad in the baseline model, now do much better in singling out ALLD players in comparison to the binary assessment model. Finally, $L8$ (Fig. 3h), which eventually judges everyone as bad under binary assessment, can judge its own kind as good much more easily when players use quantitative assessment, and accurately singles out defectors. Unconditional cooperators however are often assigned a bad reputation, due to $L8$ judging both good and bad players as bad when they cooperate with another bad individual. We note here that the reputation dynamics between $L8$, ALLD and ALLC have two very stable configurations: one where the norm judges itself and ALLC as predominantly good, and one where it judges itself as predominantly good and ALLC as bad. We have visualized the latter outcome in Figs. 2a and 3h, given that it is the slightly more frequent one. This issue does however not affect the subsequent results presented in this paper.

From this, we can see that, compared with the binary assessment common in the literature, our results so far suggest that quantitative assessment indeed can stabilize reputations in a population with fixed composition. In fact, it can correct disagreements introduced by noise, such that populations quickly recover from erroneous reputation assignments and misjudgments. We analyze this recovery in the SI, Section 1. There, we explore the effects of a single initial disagreement in a homogeneous population of leading eight players using quantitative assessment with $R = 3$. Assuming that there are no further errors or noise, we study how likely each of the eight social norms recovers,

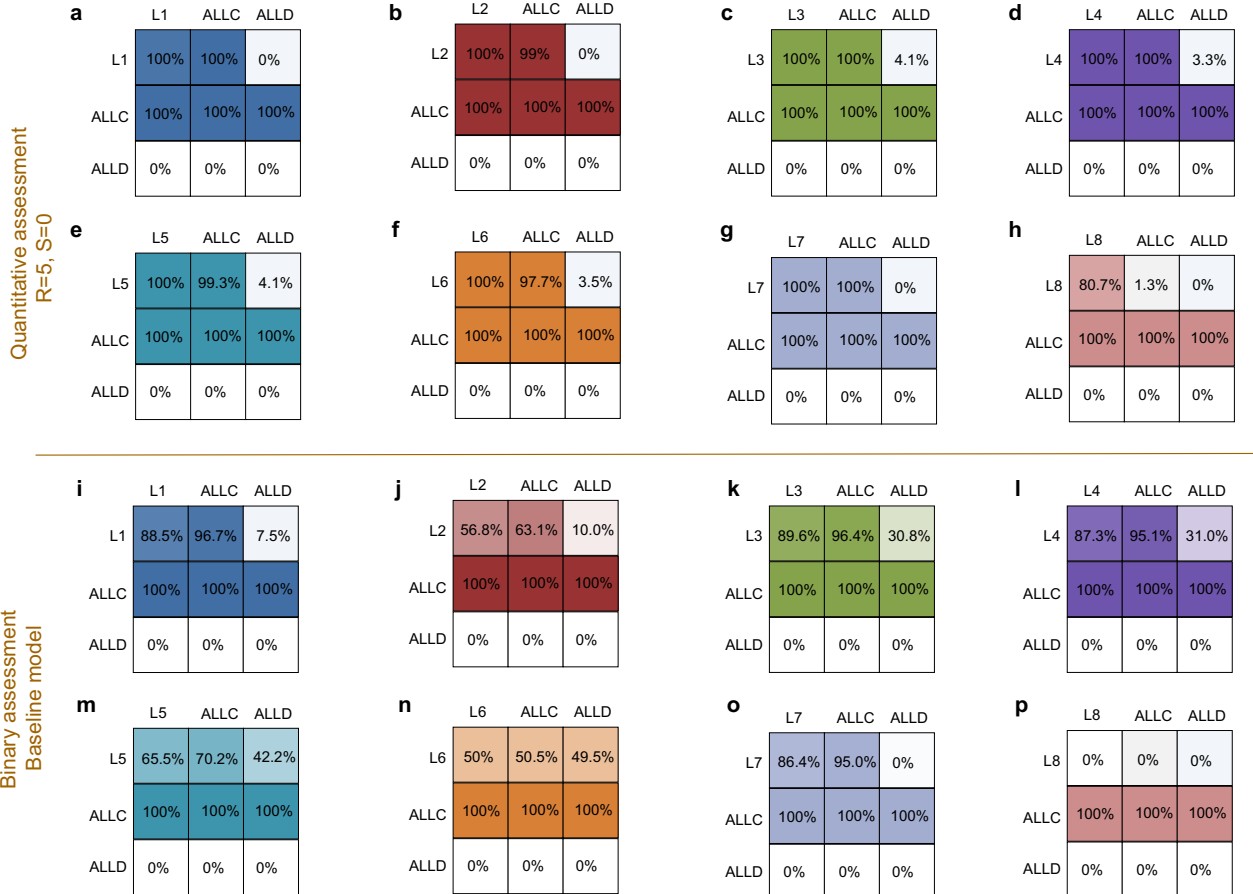

**Fig. 3 | Quantitative assessment improves the accuracy of reputation assignments by leading eight players.** We show the average overall judgments that players with the frame of reference $R$ make of each other when comparing others' reputation scores with the threshold (**a–h**). As the basis for comparison to the baseline model, we use the average images that players have of each other when they use the standard binary assessment (**i–p**). We observe that quantitative assessment and more nuanced reputations lead to a clear improvement of the accuracy with which players assign each other images. All leading eight norms achieve a perfectly correlated good self-image, as opposed to the baseline model, where only $L1$ (**i**) and $L7$ (**o**) achieve a self-image of more than 80% good. Players using quantitative assessment also do much better in judging $ALLD$ as bad, and with the exception of the (less stable) $L8$ (**h**, **p**), also manage to assess ALLC as close to 100% good. This hints at the power of a more refined reputation dynamics. The parameters are the same as in Fig. 2.

and how long this recovery takes. By modeling the reputation dynamics as a Markov chain, we analytically show that recovery probabilities and expected recovery times for all eight social norms with quantitative assessment are upper bounded by the corresponding quantities in the binary case. Hence, recovery in the quantitative setting occurs with higher probability and in fewer steps compared to the setting where reputations are binary. In fact, when we simulate the recovery process, we find that the outcomes now come close to a setting with perfect information (Fig. 4). The recovery times for all leading eight norms with quantitative assessment are linear in population size $N$. This is a stark contrast to the results for binary reputations, where recovery time can be of order $N \log N$[44].

In a next step, we now investigate how well the leading eight fare in evolution where population composition can change over time, in order to understand whether cooperation effectively emerges when players use quantitative assessment.

**Evolutionary dynamics.** We now aim to understand the effect of quantitative assessment when players' norms are not fixed, such that their abundance can change over time. We explore how likely it is for a leading eight strategy to evolve, and what cooperation rates are achieved in the population. Similar to our previous setup and to what is common in the literature on indirect reciprocity, we again consider a minimalistic scenario where players can choose from only three norms:

a leading eight norm $L_i$, $ALLD$ and $ALLC$. We assume that the evolution of social norms happens on a longer timescale that is separate from the reputation dynamics. This implies that the reputation dynamics have reached stationarity by the time that social norms change. Iterating the elementary process of reputation updating, we can not only calculate how often on average player $i$ considers player $j$ to be good, but also how often on average they cooperate with $j$. With the estimated pairwise cooperation rate $\hat{x}_{ij}$, with which player $i$ helps player $j$, we can define the payoff of player $i$ using a fixed strategy as $\pi_i = \frac{1}{N-1} \sum_{j \neq i} b\hat{x}_{ji} - c\hat{x}_{ij}$ (see "Methods" for details). To model how players adopt new strategies, we then consider simple imitation dynamics[54–56]. In every timestep of the evolutionary process, a player $i$ is picked uniformly at random to revise their norm. With probability $\mu$ they pick a random new norm. With probability $1 - \mu$, they randomly choose a role model $j$ to imitate wth a probability $P(\pi_i, \pi_j)$ depending on the difference between the two players' payoffs. This probability takes the form of the Fermi function $P(\pi_i, \pi_j) = (1 + \exp[-s(\pi_j - \pi_i)])^{-1}$. The parameter $s \geq 0$ describes the strength of selection, which measures how relevant payoffs are for updating strategies. For $s = 0$, updating happens at random, and as the parameter increases, norms with higher payoffs are more likely to be imitated. We note that for imitation processes to be a reasonable model of strategies spreading, we implicitly assume that people discuss their worldviews and moral guidelines with others.

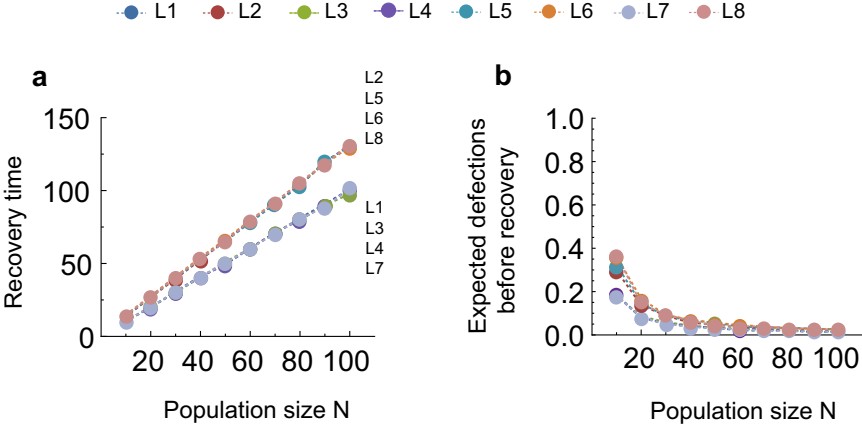

**Fig. 4 | Actual recovery times.** We simulate the recovery from the state $(0, N-2, 0)$ to a state with $s \in \{0, 1\}$ and $k + l = N - 1$ by simulating the reputation dynamics. **a** We find that the recovery time for all leading eight norms is linear in population size $N$. For $L_1, L_3, L_4, L_7$, the slope of the curve is -1, whereas the slope is -1.3 for the remaining four of the leading eight norms. **b** When we consider the average number of defections that occur before recovery, we see that this number decreases as the population size grows large. For sufficiently large populations, no defection occurs before recovery. Simulations are averaged over 10,000 rounds.

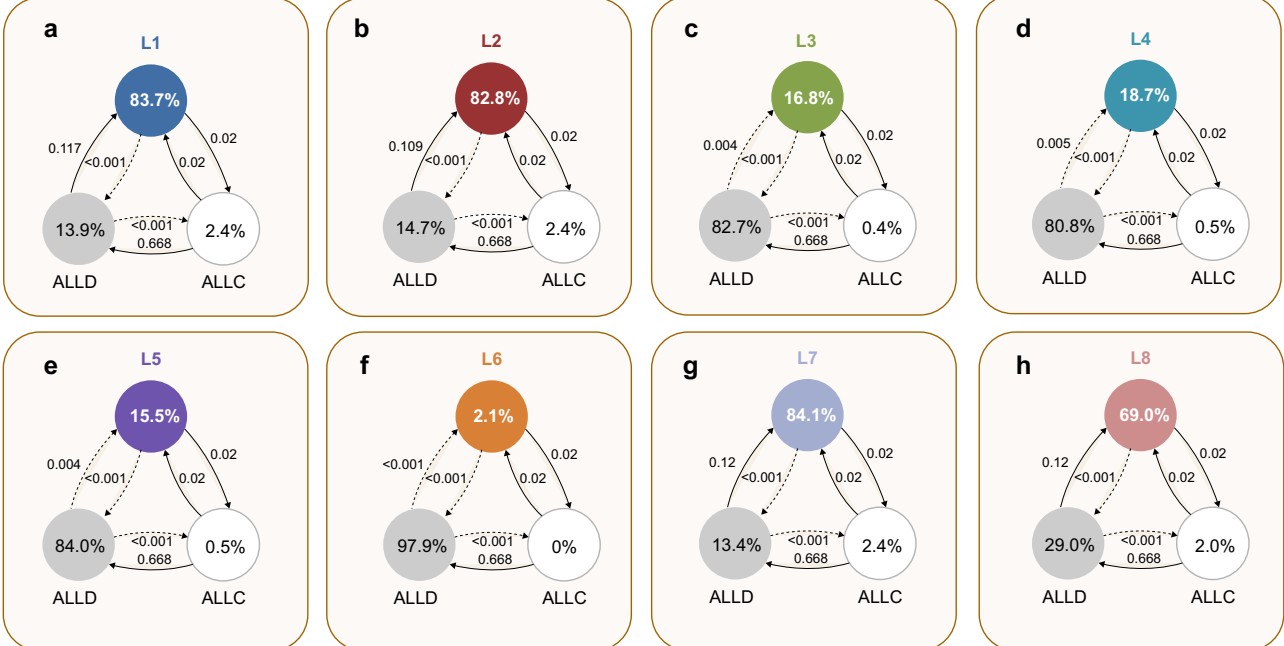

**Fig. 5 | Evolution of the leading eight with quantitative assessment.** We show the results of simulating evolutionary dynamics when players can choose among three different norms: a leading eight norm, *ALLC*, and *ALLD*. We assume that the spread of social norms is described by a pairwise comparison process[54], such that norms of players with high payoffs are more likely to be successful. Here, we use the limit of rare mutations, such that populations are homogeneous most of the time[57,58,85]. Numbers in circles show how often each social norm is adopted on average. Arrows indicate fixation probabilities, i.e., how likely it is for other social norms to invade a given resident population. Solid arrows indicate that the respective transition is more likely to occur than expected under neutrality, whereas dotted arrows indicate that the respective transition is comparably unlikely. We see that four of the eight considered norms, *L1* (**a**), *L2* (**b**), *L7* (**g**), and *L8* (**h**) achieve high abundance in equilibrium, with *L1*, *L2*, and *L7* played over 80% of the time. The remaining four norms, *L3* (**c**), *L4* (**d**), *L5* (**e**), *L6* (**f**) do not evolve in large proportions, and the respective dynamics strongly favor *ALLD*. Parameters: $R = 5$, $S = 0$, $N = 50$, $\varepsilon = 0.05$, $b = 5$, $c = 1$, $q = 0.9$, using a strength of selection of $s = 1$.

The resulting process is ergodic, due to the possibility of random mutations in every timestep. Thus, it gives rise to a unique stationary distribution, called selection–mutation equilibrium. This distribution represents the abundance of each strategy in the long run. To calculate the average cooperation rate in the population, the payoffs of the individual strategies are then weighted with this equilibrium abundance. In the following, we will assume that mutations are rare[57,58], which implies that populations are homogeneous most of the time: a new mutant only arises when the previous mutant has either gone extinct or has fixed in the population. We can calculate the fixation probability of a mutant into a resident population with social norm $R$ explicitly[59].

Figure 5 visualizes the evolutionary dynamics between each leading eight strategy, *ALLC* and *ALLD*. We find that in four cases, for $L3 - L6$ (Fig. 5c–f), the leading eight norm does not evolve. Once players have learned to use *ALLD*, there is very little chance of reestablishing cooperation. On the other hand, three of the leading eight, $L1, L2, L7$ (Fig. 5a, b, g) are more than 80% abundant in equilibrium, which means that their evolution is strongly favored. $L8$ is also played almost 70% over the course of evolution. This stands in stark contrast

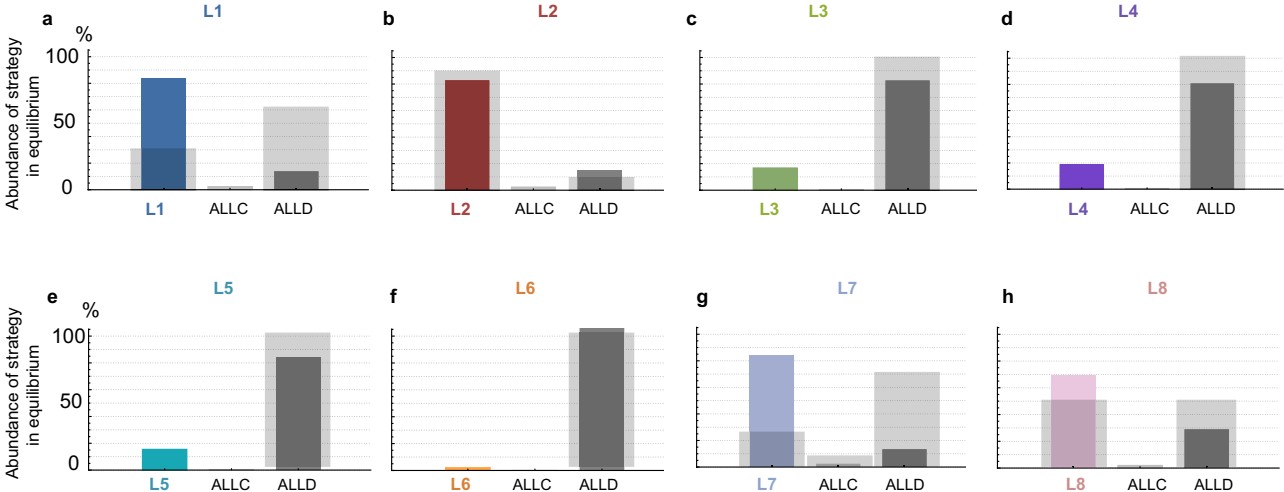

**Fig. 6 | Four of the leading eight evolve in high proportions for quantitative assessment.** We compare the abundance of the leading eight strategies in selection–mutation equilibrium between the case of quantitative assessment and the baseline model. We use the same evolutionary process and setup as in Fig. 5 and present the changes in how often each norm is played on average. Colored bars represent the abundance in equilibrium under quantitative assessment, while the light gray bars in the background of each panel represent the results in the baseline model. We find that four of the eight strategies now evolve much more readily (**a**, **b**, **g**, **h**) than in the baseline model, and are played in large proportions. The three remaining strategies (**c**, **d**, **e**, **f**), which do not evolve at all in the baseline model, only do slightly better due to still being outcompeted by *ALLD*. The parameters are the same as in Fig. 5.

to the baseline model, where only *L*2 achieves an equilibrium abundance of that magnitude (Fig. 6b), and where most leading eight strategies do not evolve in meaningful proportions. We can see that this difference is mainly due to sufficient probability of *L*1, *L*2, *L*7, and *L*8 invading *ALLD* (Fig. 5a–c, g). *ALLC* and each leading eight norm are approximately neutral with respect to each other (their fixation probability into one another is ≈1/*N* each), and *ALLC* is invaded by *ALLD*. Thus, the fixation probability of the leading eight norm into *ALLD* becomes the deciding factor. This also suggests that the power of quantitative assessment lies mainly in its ability to correct errors in the norm's self-image by effectively reducing noise in the population, hence enabling a more accurate judgment of defectors. We can see this in the behavior of *L*8 (Fig. 6h), which is less abundant compared to e.g., *L*7 due to the fact that it does not always judge itself as good in the presence of ALLD. Meanwhile, *L*3, *L*4, *L*5, *L*6 are unsuccessful in evolution, as they have difficulties labeling defectors as bad already in a non-noisy environment, and therefore do not profit as much from the error-correcting quality of quantitative assessment. This shows that for a higher-order norm to be stable under noisy and private information, it must at least negatively assess a bad player defecting against another bad player. In the SI Section 2, we detail this point in an explicit characterization of the properties that are necessary for a norm to be successful.

Figure 7 then visualizes how the eight norms' abundance in equilibrium translates into cooperation rates. We see that compared to the baseline model (Fig. 7d–f), seven of eight norms lead to increased cooperation, with only *L*6 completely failing to evolve cooperative behavior. *L*1, *L*2, *L*7 give rates of almost 90%, whereas *L*8, which leads to no cooperation at all in the baseline model, also is boosted. These findings are highly robust when we vary parameters (Fig. 7a–c), most remarkably even when we vary the error rate (Fig. 7a). Even at a noise level of $\varepsilon = 0.1$, cooperation evolves and is maintained at over 80% for *L*1, *L*2, *L*7, while it also does not fall below 60% for *L*8. This is due to both the high abundance of these four leading eight norms when players use quantitative assessment, and to an increased self-cooperation rate in homogeneous populations even when errors are more frequent. Quantitative assessment thus changes the behavior of the leading eight norms' cooperation rates significantly.

Finally, we explore the effect of the size of the assessment scale, i.e., the range of the reputation scores (Fig. 8). The case of two possible ranks (reputation values) corresponds to binary assessment. Notably, we find that for *L*1, *L*2, *L*7, *L*8, cooperation rates change non-monotonically in the number of possible reputation scores (Fig. 8a). In fact, choosing an assessment scale smaller than taking $R = 5$, as previously used, leads to even more cooperation, especially for the more unstable *L*8 with its strict assessment rule. Figure 8b suggests that an intermediate number of levels gives the four successful social norms a higher abundance, due to a higher probability of invading *ALLD*. The norm *L*2 is also a special case: here, going from binary assessment to quantitative assessment with $R = 1$ harms the cooperation rate, as abundance drops down to only 46% from 89%. These observations suggest that there is a tradeoff between the size of the assessment scale and the increased self-cooperation rate going towards 100% that comes with more fine-grained reputation scores (Fig. 8c). Intuitively, an increased range of reputation scores can act as a buffer that not only protects good reputations from misjudgments, but can also make it harder to quickly correct mistakes. In turn, this can lead to sunk costs when players cooperate too often with defectors. This becomes a particular issue for *L*2, as it is the only norm to actually evolve in a large proportion in the baseline model, but also is sensitive due to its negative assessment of good players' cooperation with bad players. Therefore, the net effect is negative when going from binary assessment to quantitative assessment with $R = 1$. The other norms do not exhibit this drop in cooperation rates when moving to $R = 1$. They profit far more from the increased error correction, such that the negative effect of larger reputation intervals only becomes apparent later on.

This shows that changing some of the inner workings of the reputation updating has a clearly pronounced effect on the overall dynamics by effectively minimizing the effect of noise. In contrast, however, we find that merely changing the threshold $S$ for a player to be overall judged as "good", while keeping the frame of reference constant, does not have the same impact (Fig. 9).

## Discussion

Indirect reciprocity explores how people form good reputations, and how social norms evolve[2–4]. It approaches fundamental moral

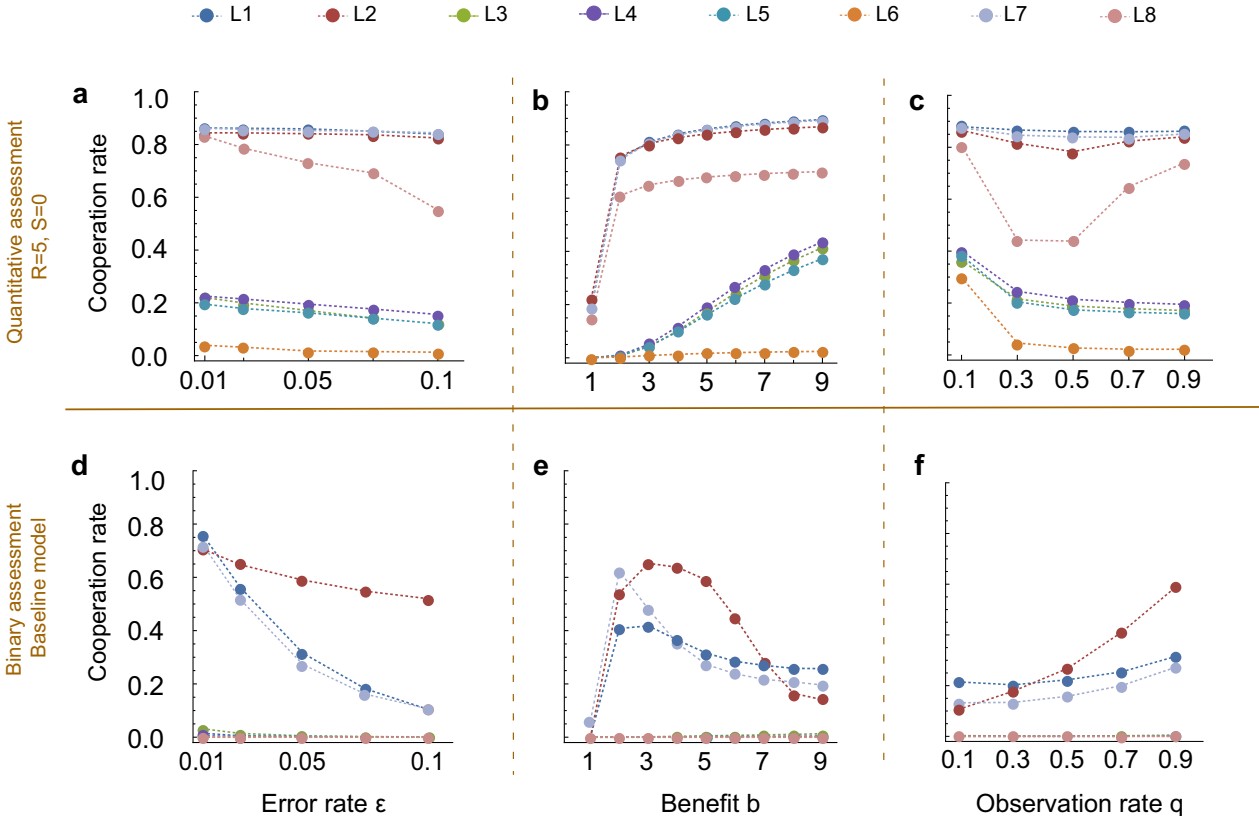

**Fig. 7 | Quantitative assessment has a positive impact on cooperation rates.** We vary the noise on observations $\varepsilon$, the benefit-to-cost ratio $b/c$, with $c=1$, and observation probability $q$. All other parameters remain constant at the values of Fig. 5. In each scenario, we plot the average cooperation rate of each individual leading eight norm when they compete against *ALLD* and ALLC, according to the selection–mutation equilibrium of the evolutionary process. We can compare the results when players use refined assessment with $R=5$ (**a–c**) with the outcome of the binary assessment in the baseline model[44] (**d–f**). **a** Under quantitative

assessment, cooperation rates of *L1*, *L2*, and *L7* remain at around 85% even when the error rate $\varepsilon$ increases to 0.1. The generally more unstable *L8* is more affected by the increased noise, but still remains above 50% even at $\varepsilon=0.1$. **b** Increasing the benefit of cooperation $b$ leads to an increase in cooperation rate for all eight considered norms in contrast to the baseline. **c** When we increase the observation probability $q$, the behavior of the leading eight norms' cooperation rates is also markedly different from the baseline. *L1*, *L7* are barely affected while *L2* and *L8* exhibit nonlinearity for intermediate values of $q$.

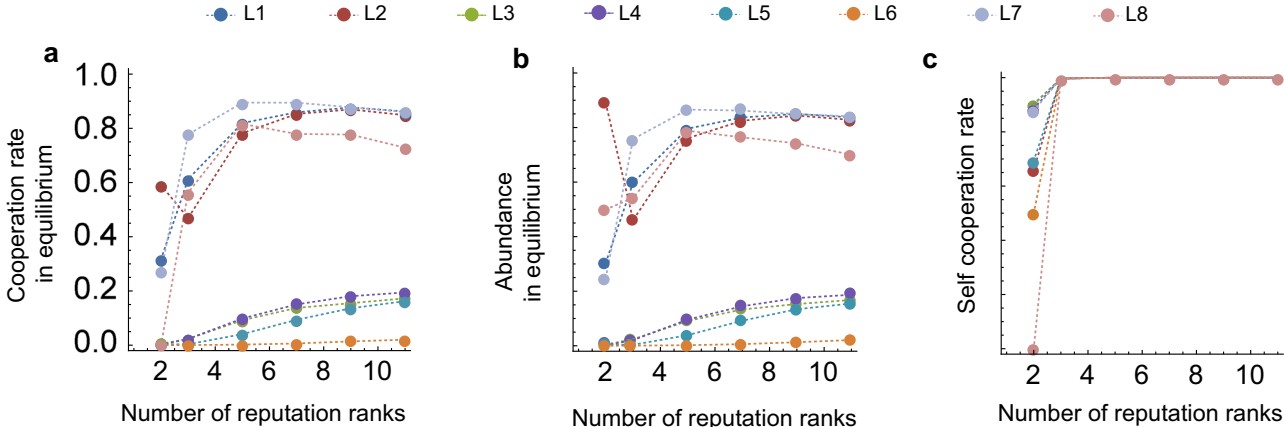

**Fig. 8 | Varying the frame of reference for quantitative assessment.** For this figure, we repeat the evolutionary simulations shown in Fig. 5, and vary the frame of reference $R$. That is, we explore the impact of the number of possible reputation ranks on cooperation, including the case of binary assessment with two reputation ranks. **a** We show the cooperation rate in equilibrium for the leading eight norms as the number of reputation ranks increases. We note that for the four successful norms *L1*, *L2*, *L7*, *L8*, the largest frame of reference does not correspond to the highest cooperation rate. An intermediate number of

ranks is the most beneficial. *L2* also exhibits a drop in cooperation rate from binary assessment to $R=1$ (i.e., 3 reputation ranks). The behavior of the cooperation rates is mainly determined by the behavior of the equilibrium abundance of the eight norms as the frame of reference varies (**b**). Meanwhile, self-cooperation rates quickly increase to 1 as the frame of reference increases (**c**), which implies that the leading eight players have a perfectly correlated image of each other once assessment is more nuanced. The parameters are the same as in Fig. 5.

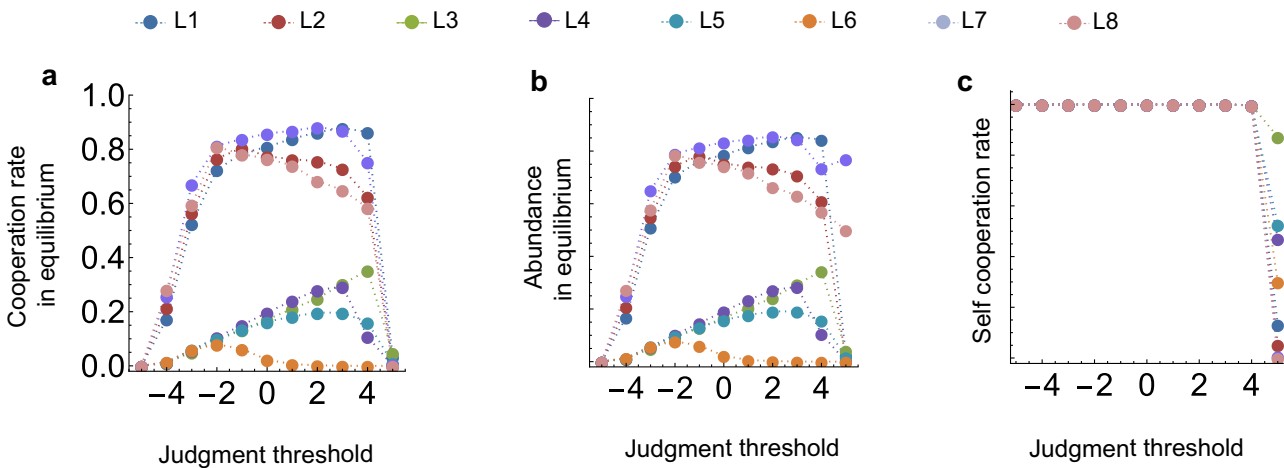

**Fig. 9 | Varying the threshold for an overall good reputation of players.** For this figure, we repeat the evolutionary simulations shown in Fig. 5, and vary the threshold $S$ for an overall good judgment of a player. **a, b** We find that cooperation rates and equilibrium abundance of the leading eight norms are not significantly different for threshold values between $S = -2$ and $S = 4$. Additionally, we observe that values of $S$ closer to the value $-R$ (here, $R = 5$) are more detrimental than those closer to $+R$; having a very large buffer for negative reputations thus seems to be less of an issue than having a very large buffer for positive reputations. **c** Self-cooperation rates are not affected by a change in threshold and stay at a value of 1, except for the value $S = R$, where self-cooperation drops. The parameters are the same as in Fig. 5.

questions, for example, how a good person should act and how goodness should be defined in the first place, from the viewpoint of evolutionary biology[1]. A crucial question in indirect reciprocity is how social norms can sustain cooperation when individuals judge each other according to their own moral standards and idea of goodness[13,14,33]. Previous work suggests that social norms can only reliably maintain cooperative behavior if they are sufficiently complex. Simple rules like Image Scoring, which tie the idea of "good" and "bad" individuals merely to those individuals' actions, have been shown to be unstable[29–31]. Instead, Ohtsuki and Iwasa proposed the so-called "leading eight" social norms[28,33]. These norms contextualize actions by taking into account the reputations of the respective interaction partners. In this way, they are able to differentiate between justified and unjustified defections. Yet these leading eight norms are susceptible to the effects of private and noisy information[44]. When individuals make assessments based on their own private observations, populations may get fragmented into several subgroups with no or little cooperation between them[45].

Here, we show that incorporating more nuanced opinions, which are a natural feature of real-life reputation systems, in a model of indirect reciprocity results in a more positive outcome. To this end, we consider the leading eight with a more refined assessment system. Individuals privately keep track of others' reputation in the form of integer scores. Positive interactions translate into a score increment, whereas negative interactions reduce the respective individual's score. This implies that "goodness" is reinterpreted in a broader way. Under binary assessment, there is only one "good" and one "bad" label. Instead, in our model, a person's goodness comes in several shades. Similar ideas have also been proposed to make 'Image Scoring' less vulnerable to errors[26,48,49]. Our simulations and analytical results show that this error-correcting capability clearly works in favor of the leading eight strategies. Since quantitative assessments can act as a buffer for misjudgments due to noise and lapses in observations, minor disagreements between individuals are no longer the same threat to cooperation. At the same time, consistent defectors have their reputation score further decreased (as much as the range of reputation scores allows), which further strengthens the stability of cooperative norms.

Ohtsuki and Iwasa's landmark papers identified certain properties of a higher-order norm that need to be fulfilled for a norm to be successful. For example, to maintain cooperation, a good player who cooperates with another good player ought to keep its good reputation. Furthermore, defectors must be recognized, meaning that defection against a good player must be assessed as bad. In addition, there need to be ways to punish defectors. In particular, good individuals who defect against bad population members should maintain their good reputation. Finally, forgiveness must be possible: the cooperation of a bad player with a good player should restore their reputation. All other bits of the norms are left flexible, resulting in the leading eight norms. We meanwhile show that in the presence of private and noisy information, one more criterion should be added to this characterization: norms must not be 'gullible': they should never assess a bad player defecting against another bad player as good (making them easily deceived by false shows of solidarity). This requirement determines another bit of the assessment rule (see SI, Section 2), further reducing the number of successful social norms from eight to four.

From a psychology perspective, it might come as little surprise that nuanced reputations and tolerance for a few negative experiences with others help to resolve disagreements. So-called dichotomous thinking[60]— thinking in simple terms of binary opposition instead of seeing shades of gray—is assumed to be beneficial for quick decision-making and taking control of situations, but at the same time has been found to be a cognitive distortion correlated with personality disorders[61–64]. This bias is particularly prevalent in Cluster B and C disorders such as borderline personality disorder and narcissistic personality disorder, which are known for destructive tendencies in interpersonal relationships and/or difficulties maintaining bonds with others[65–67]. Dichotomous thinking also has been studied as a hallmark bias of traits of antisocial behavior[68]. Particularly simplistic worldviews with no tolerance for single missteps ("You are either with us or against us") can be harmful on a larger scale, which can be seen in extreme political partisanship[69] and the legislatory deadlocks and social fractures it can evoke.

Another significant feature of our results is that an intermediate value of reputation ranges is optimal for the evolution of cooperation. Reputations should not be measured too coarsely, nor should they be too fine-grained. This observation may be related to previous arguments suggesting that limits on human information processing capability make it difficult for an observer to use too many categories when making judgments[70]. In fact, various empirical investigations of optimal rating scales in marketing research and beyond have found

that increasing the number of response categories beyond a value over 10 at most holds little benefit[71–73]. Other work has described a tradeoff between a too-coarse and too fine-grained rating scale[74], mirroring our results.

Furthermore, we can weigh our results against empirical studies on indirect reciprocity and general "social status" that gives information over more than the last round of interactions[75–78]. These studies support the conclusion that when players know enough about their partners' previous decisions, they are likely to engage in indirect reciprocity. Players' likelihood of cooperation then depends on their partner's cumulative status. How exactly this status is translated into behavior is encoded in an individual's specific norm. With our study, we go one step further in describing a social status that is formed by players' own observations and potentially more complex assessments in each round, i.e., a status that is not simply given as a piece of information to take into account. We show that when community members privately keep track of others' reputation scores by accumulating observations and assessments of others' actions according to higher-order norms, indirect reciprocity can in fact stabilize cooperation even when information is noisy. Further experimental studies can provide an underpinning for our results by demonstrating that second-order information can enhance indirect reciprocity even when cooperation is costly and therefore hard to maintain[4]. Players in fact seek out higher-order information, even when they can already see their interaction partners' recent decisions over multiple rounds[79].

Here, we have studied the reputation dynamics in well-mixed populations in which a resident social norm is regularly challenged by alternative social norms. Interesting additional effects may occur, for example, when several social norms compete simultaneously, or when we take into account population structure. One can assume that the topology or group structure underlying a population[80–83] can play a significant role in the question of how well quantitative assessment can diminish the effect of errors. For sparse topologies, it is not trivial to answer this question, as too much buffer for negative experiences can become a hindrance when a player cannot observe certain actions at all. These limitations notwithstanding, our present findings demonstrate how modeling a reputation system to be more nuanced gives rise to a clear increase in cooperation rates when individuals' behavior is governed by the leading eight social norms. We in fact show that (some of) these norms then continue to maintain cooperation, even when information is private, noisy, and imperfect, if only reputations are sufficiently fine-grained. Overall, our study thus highlights how broader definitions of goodness can benefit a community by helping them to maintain positive social relations.

## Methods
### Reputation dynamics
As a first step, we consider how the reputations of players change over time. For this analysis, we assume that social norms are fixed for each player. We use the image matrix $M(t)$ to record the reputation scores that players assign to one another. This matrix is time-dependent and is updated in every step of the dynamics. An entry $r_{ij}, r \in [-R, R]$ means that player $j$ has score $r_{ij}$ in the eyes of player $i$. These scores translate into overall judgments that individuals make of their co-players: to label (i.e., judge) a co-player $j$ as good or bad, an individual $i$ compares the respective score $r_{ij}$ with a threshold $S$. If $r_{ij} < S$, player $i$ labels player $j$ as "bad". If $r_{ij} \geq S$, player $i$ judges player $j$ as "good". In the following, we will denote the overall judgment of a player based on the score $r_{ij}$ as $J(r_{ij}) = J_{ij}$, with $J_{ij} \in [G, B]$. Before the first round, we assume that all entries are $r_{ij} = 0$, i.e., that all players have a good label of each other regardless of their social norm, but assign the lowest possible "good" reputation score. Thereafter, in every round, two players are chosen at random to act as donors and recipients. The donor can confer a benefit $b$ to the recipient at own cost $c$. This action depends on the donor's norm, and additionally on the label they assign to themselves and the

recipient, i.e., the judgment based on the corresponding reputation scores. Every other co-player can observe the action with probability $q$, and these observations are individually subject to misperception, which happens with probability $\varepsilon$. Observers as well as the donor and recipient then update the donor's reputation score. This happens according to their assessment rule and the labels they currently assign to the interaction partners, based on each partner's score. If player $i$ assesses player $j$'s action to be good, the reputation score $r_{ij}(t)$ will increase to $r_{ij}(t+1) = r_{ij}(t) + 1$. If they assess player $j$'s action to be bad, the reputation score will decrease to $r_{ij}(t+1) = r_{ij}(t) - 1$. These score updates are recorded in the image matrix $M(t+1)$.

We can iterate this process over many rounds and calculate the average reputation scores, and more importantly, binary labels that players assign to each other. Specifically, if players have interacted for $T$ rounds in total, the average label that player $i$'s assigns to player $j$ is defined as $\frac{1}{T}\sum_{t=1}^{T} J_{ij}(t)$. Additionally, our simulations let us estimate the pairwise cooperation rate of $i$ against $j$, i.e., how often player $i$ cooperates with $j$ on average. Denoting this cooperation rate by $\hat{x}_{ij}$, we can then calculate player $i$'s payoff for a fixed social norm as

$$\pi_i = \frac{1}{N-1}\sum_{j \neq i} b\hat{x}_{ji} - c\hat{x}_{ij}. \tag{1}$$

That is, the average payoff is the result of the benefits gained by $i$ through $j$ cooperating ($b\hat{x}_{ji}$), reduced by the costs of $i$'s own cooperation with $j$ ($c\hat{x}_{ij}$), averaged over all of $i$'s $N-1$ co-players.

We illustrate our approach in Figs. 2 and 3, where we consider a population consisting of equal parts *ALLD*, *ALLC*, and a leading eight norm. In Fig. 2, we present snapshots of the image matrix at time $T = 2 \times 10^6$. For each of the leading eight strategies, we show both a visualization of the reputation scores as well as a visualization of the overall judgments $J_{ij}$, i.e., the labels players assign to each other based on the reputation scores. In Fig. 3, we show the respective average overall judgments for both the quantitative assessment model with frame of reference $R = 5$ and the baseline model wth binary assessment.

We note that in the case of *L8*, there are two relatively stable configurations when we consider reputation dynamics in a population that also includes *ALLD* and *ALLC* players. Since *L8* judges players who cooperate with bad co-players as bad themselves, they often tend to judge unconditional cooperators as bad once they meet unconditional defectors. Therefore, the two stable configurations are one where *L8* judges itself and *ALLC* as predominantly good, and one where it judges itself as predominantly good, but *ALLC* as predominantly bad. In Figs. 2 and 3, we show the second scenario, since it appears slightly more frequently in our simulations. However, this issue does not affect our other results, since in this work we do not consider the coexistence of more than two strategies in the population for our subsequent analysis.

To explore the robustness of our results, we have additionally run simulations where players start with a negative reputation score, i.e., a bad label. We obtain the same result as in Fig. 2, except for *L7* and *L8*: If there are no players with a good label to begin with, it is impossible for donors to gain in reputation. Hence, all *L7* and *L8* players keep assigning negative scores to all co-players under such an initial condition.

The assumption of a symmetric interval of possible scores, $[-R, R]$ with the threshold at $S = 0$ naturally implies that there are slightly more possible ranks for a "good" player than a "bad" one. A player with $r = S$ will be judged as "good". To test this bias, we have also run simulations where a player with $r = S$ will be judged as "bad", and have found no significant qualitative difference in results as long as players start with a good reputation.

We can naturally extend our analysis technique beyond the restricted setup where only three social norms compete. Our approach

can be expanded by allowing population members to choose among additional norms. However, computational complexity increases rapidly in the number of considered social norms. Additionally, our presented results are easy to compare to previous work using very similar setups. They capture the crucial feature that disagreements between players using a leading eight norm are drastically reduced with quantitative assessment.

## Evolutionary dynamics

In the next step, we explore a setting where individuals' social norms are no longer fixed. We analyze which norms players themselves choose to adopt over time, following the methods in previous work[44,56]. For this analysis, we assume that players change their norms over a timescale that is longer and separate from the timescale of the reputation dynamics with fixed norms. To formally describe the process on this longer time scale, we assume that individuals adopt new social norms based on pairwise comparison[54]. In every evolutionary timestep, one player is randomly chosen from the population. With probability $\mu$, with $\mu$ the mutation rate, this player picks a new social norm at random from the respective set of available norms. Meanwhile, with $1 - \mu$, the focal player randomly chooses a role model from the population. If the focal player's payoff according to Eq. (1) is given by $\pi_i$ and the role model's payoff is $\pi_j$, then the focal player adopts the role model's norm with probability $P(\pi_i, \pi_j) = (1 + \exp[-s(\pi_j - \pi_i)])^{-1}$. The parameter $s$ is called the strength of selection. When $s$ is small, imitation occurs largely at random. For larger $s$, however, players are most likely to imitate those role models with a higher payoff.

In this work, players can choose from three social norms, a given leading eight norm, ALLC, and ALLD. We use this setup to keep the system as simple as possible, while also following the lead of much of the previous work in the field[25,30,31,36,49,51,84].

In evolutionary game theory, imitation processes are often used as a standard model to describe the spread of strategies in a population. For this model to be reasonable, it is necessary to assume that players are able to infer their co-players' strategies from their observed behaviors, which can be difficult in indirect reciprocity. We therefore implicitly assume in this work that people discuss their worldviews and moral guidelines with others. Instead of imitation, one could in contrast also consider an alternative model by assuming that social norms spread through a birth-death process, i.e., that parents pass on their own social norms to their children. However, for the function we use to model imitation, the imitation process is equivalent to a birth-death process with exponential fitness mapping[55]. Thus, our results would not change.

This evolutionary process based on mutations and imitation is ergodic on the space of all possible population compositions. Hence, it gives rise to a unique stationary distribution, which we refer to as the selection–mutation equilibrium. This equilibrium reflects how often each of the available social norms is adopted over time. To be able to efficiently calculate exact strategy abundances in this work, we use the limit of rare mutations, which assumes that populations are homogeneous most of the time. When a mutation arises, it either fixes in the population or goes extinct before the next mutant appears. We can calculate this fixation probability of a mutant with social norm $M$ into a resident population with social norm $R$ explicitly as[59]

$$\rho_{MR} = \frac{1}{1 + \sum_{i=1}^{n-1} \prod_{k=1}^{i} e^{-\beta(\pi_M(k) - \pi_R(k))}}. \qquad (2)$$

Here, $\pi_M(k)$ and $\pi_R(k)$ are the respective payoffs of mutants ($M$) and residents ($R$) when $k$ individuals in the population employ the mutant norm. This means that we can describe the evolution of the social norms between three available norms in the rare mutation limit as a Markov chain with three states. These three states correspond to the respective homogeneous populations, i.e., all players using ALLC,

all using ALLD or all players using a leading eight norm. We note that this feature of the rare mutation limit is the reason why our evolutionary results are not affected by the stability issues encountered in the reputation dynamics between L8, ALLC and ALLD. Given the pairwise fixation probabilities according to Eq. (2), the respective transition matrix of this evolutionary Markov chain is given by

$$W = \begin{pmatrix} 1 - \frac{1}{2}(\rho_{LC} + \rho_{LD}) & \frac{1}{2}\rho_{LC} & \frac{1}{2}\rho_{LD} \\ \frac{1}{2}\rho_{CL} & 1 - \frac{1}{2}(\rho_{CL} + \rho_{CD}) & \frac{1}{2}\rho_{CD} \\ \frac{1}{2}\rho_{DL} & \frac{1}{2}\rho_{DC} & 1 - \frac{1}{2}(\rho_{DL} + \rho_{DC}) \end{pmatrix}. \qquad (3)$$

The stationary distribution of this transition matrix is the selection--equilibrium of the process for rare mutations[57]. Given this equilibrium, we can compute how often players cooperate on average by taking the average cooperation rate of each homogeneous population, and multiplying it by how often we are to observe the respective homogeneous population in equilibrium.

We use this approach in Figs. 7, 8, and 6, where we first simulated the reputation dynamics for all possible population compositions, $(n_L, n_C, n_D)$, with $N = n_L + n_C + n_D = 50$ and $5 \times 10^6$ steps each. Here, $n_L$ stands for the number of players using a leading eight norm, $n_C$ for the number of players using ALLC, and $n_D$ for the number of ALLD players. Payoffs are computed with Eq. (1), as explained in the subsection on the reputation dynamics.

**Specific methods employed for the figures.** Figure 2 shows the results of reputation dynamics in a population of $N = 90$ players. The population composition is as follows: 1/3 uses ALLD, 1/3 uses ALLC, and 1/3 uses a leading eight norm. We consider the leading eight norms with quantitative assessment of reference frame $R = 5$, i.e., reputation scores in the interval $[-5, 5]$ (Fig. 2c–j). The threshold used for computing the overall judgment is $S = 0$. We assume that information is noisy ($\varepsilon = 0.05$ and $q = 0.9$). Snapshots are taken at $T = 2 \times 10^6$.

Figure 3 shows the average overall judgments in a population of the same composition as in Fig. 2a–h. In Fig. 3i–p, we run the same simulation for a population whose leading eight players use binary assessment instead. All parameters are the same as in Fig. 2.

For Fig. 4, we simulate reputation dynamics in a homogeneous population of leading eight players using quantitative assessment ($R = 3, S = 0$) when information is perfect $\varepsilon = 0, q = 1$. With this, we simulate the recovery time from a single disagreement (a, as well as the average number of defections until recovery (b), while varying population size $N$.

In Fig. 5, we show the abundance of ALLC, ALLD and each leading eight norm with quantitative assessment in the selection–mutation equilibrium. Assessment parameters $S$ and $R$ remain as in Fig. 2. Other parameters are $b = 5, c = 1, \varepsilon = 0.05, q = 0.9$ and selection strength $s = 1$.

Figure 6 compares the equilibrium results of the evolutionary simulations in Fig. 5 with the corresponding results in the baseline model of binary assessment with $S = 1$ and possible reputation scores [0, 1]. All parameters as in Fig. 5.

In Fig. 7, we explore the effect of quantitative assessment with $R = 5, S = 0$ on cooperation in the selection–mutation equilibrium for all leading eight norms when we vary benefit $b$, noise $\varepsilon$ and observation probability $q$. We can then compare the baseline model (Fig. 7d–f) using binary assessment ($S = 1$, with possible reputation scores [0, 1]) with our quantitative assessment model (Fig. 7a–c). The other parameters remain as in Fig. 5.

For Fig. 8, we have repeated the evolutionary simulations to explore the effect of varying the frame of reference. We consider cooperation rate in the selection–mutation equilibrium (a), abundance in equilibrium (Fig. 8b), and the self-cooperation rate in a homogeneous population of leading eight players (Fig. 8c). Other parameters are the same as in Fig. 5.

For Fig. 9, we proceed as in Fig. 8, but change the threshold $S$ for a "good" overall reputation, while keeping the number of reputation levels constant ($R = 5$). In each simulation, players start with a reputation score equal to the threshold value $S$. All other parameters are the same as in Fig. 5.

## Reporting summary
Further information on research design is available in the Nature Portfolio Reporting Summary linked to this article.

## Data availability
There are no empirical data associated with this study. Data for the main text was generated with Python 2.7 and visualized with Mathematica 11. The scripts used to generate all data are available online at https://osf.io/n35ah/?view_only=35b55d71e6ab46219fc40b2a32639152.

## Code availability
All simulations and numerical calculations have been performed with Python 2.7, and the generated data was visualized with Mathematica 11. The Python script used to simulate the reputation dynamics and calculate the selection–mutation equilibrium and average cooperation rates, as well as the script simulating the recovery process from a single disagreement, are available online at https://osf.io/n35ah/?view_only=35b55d71e6ab46219fc40b2a32639152.

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

## Acknowledgements
This work was supported by the European Research Council CoG 863818 (ForM-SMArt) (to K.C.) and the European Research Council Starting Grant 850529: E-DIRECT (to C.H.). L.S. received additional partial support by the Austrian Science Fund (FWF) under grant Z211-N23 (Wittgenstein Award), and also thanks the support by the Stochastic Analysis and Application Research Center (SAARC) under National Research Foundation of Korea grant NRF-2019R1A5A1028324. The authors additionally thank Stefan Schmid for providing access to his lab infrastructure at the University of Vienna for the purpose of collecting simulation data.

## Author contributions
All authors conceived and discussed the study during the internship of F.E. at IST Austria. L.S. collected the data, analyzed it, and wrote the first draft of the manuscript. L.S., C.H., and K.C. discussed the results and edited the manuscript. All authors reviewed the manuscript.

## Competing interests
The authors declare no competing interests.
