## [Peer Review File · Nature Communications]

Quantitative assessment can stabilize indirect reciprocity under imperfect informationREVIEWER COMMENTS

Reviewer #1 (Remarks to the Author):

This manuscript considers a theoretical analysis of indirect reciprocity under imperfect information. Compared to studies on complete information or public assessment, there are not many studies that deal with such imperfect information. Therefore, the suitable norms under this system have not yet been fully analyzed and determined. In particular, there are some competing results under binary reputation. To add effective evidence to the argument, this manuscript attempted a systematic approach where individuals assign more nuanced (multiple rather than binary) reputations. Their agent-based simulations and theoretical analysis show that such multiple reputations have strong error-correcting properties. Thus, cooperation is resilient if the population adopts any of four norms of the leading eight, even if the information is private and noisy.

This manuscript is an excellent work that provides substantive results for an imperfectly informed version of indirect reciprocity research. The current version meets publication level in many aspects including literature review, motivational and research questions, methods, results, discussion, and presentation. I recommend its publication as it is with my confidence as an expert in the field.

Reviewer #2 (Remarks to the Author):

This paper explores indirect reciprocity in the presence of quantitative assessment of reputations, where "quantitative assessment" here means that reputations are coded on an integer scale rather than via a simple binary of "good" or "bad". The scale is then converted into a reputation via a threshold. The authors find that this type of quantitative assessment can improve the resilience of four of the leading eight social norms to invasion by unconditional defectors or cooperators, even when reputation information is private and noisy. In particular, the use of quantitative assessment helps improve the resilience of norms of indirect reciprocity by providing a mechanism for rapid error correction.

This paper contains several interesting results that will be useful to those studying models of indirect reciprocity. Whether these results will be of wide interest, i.e. of interest to the readership of Nature Communications, is not clear. I personally find the technical details of the model very interesting. However the main impact of the results presented here is to show that imperfect information isn't a problem if reputations are quantitative. And so a model of indirect reciprocity fixes a problem with a previous model of indirect reciprocity. The main justification for publishing these models in high impact journals seems to be that they contradict the findings of previous versions of the same model, also published in high impact journals. The snake eats its own tail. Why should someone not working on this specific type of model care about these findings? The authors make some efforts in this direction in the discussion (e.g. line 338-348) but it is not explored in depth, and in any case, the next iteration of the indirect reciprocity model may well show something different. I see enormous value this type of work, but fundamentally this is a technical paper about an extension of a previously published mathematical model.

Reviewer #3 (Remarks to the Author):

Research context: The paper explores how cooperative behavior can be sustained in the framework of indirect reciprocity in the absence of perfect information. Direct and indirect reciprocity are the two classical models in game theory to study social dilemmas. Indirect reciprocity is based on developing social norms of reputation, dependent on indirect interactions, that can sustain cooperation in social dilemmas.

Previous results: There has been extensive research in the area of indirect reciprocity. A fundamental theoretical result from Ohtsuki-Iwasa characterizes third-order strategies that can sustain cooperation in the ideal scenario when the reputation information is precise and perfectly synchronized information. These strategies are called leading-eight strategies. However, it has been shown in a PNAS paper that in the realistic scenario when there is noisy information or imperfect information, then the leading-eight strategies either do not evolve or are not able to sustain cooperation. Thus for the realistic scenario of indirect reciprocity with imperfect information mechanisms to sustain cooperative behavior is an important problem, which this work addresses.

Summary of results: The main result of the paper shows that while classical leading-eight strategies consider binary reputations, generalizing them to have quantitative reputations help to sustain cooperation in indirect reciprocity with imperfect information. Also surprisingly quantitative reputation does not help all leading eight strategies, but only four of the leading eight strategies.

Methods: The methods of the paper include simulation results for (a) fixation probability (Figure 3), (b) cooperation rates (Figure 4), and (c) the impact of varying reputation levels (Figure 5). The simulation results are complemented with two analytical results: (a) recovery time analysis for errors, and (b) a nice extension of Ohtsuki-Iwasa result for characterization of strategies in presence of noise which identifies four of the leading-eight strategies as useful.

Importance and novelty: I consider these results very important and appealing as they present a significant advancement of obtaining a mechanism to sustain cooperation for the fundamental model of indirect reciprocity with imperfect information. The results are also novel as the mechanism proposed is very natural and established with simulation and analytical results.

The manuscript is clearly written and presents new fundamental results which are comprehensive and novel. Hence I recommend publication in Nature Communications.

Suggestion for improvements for revision:

It would be useful if there is a figure to illustrate the role of varying the threshold for cooperation. The authors show the role of varying reputation levels (Figure 5), and it would be useful to know whether varying the threshold level for cooperation also impacts the cooperation rate.

Apart from this, I am happy to congratulate the authors to excellent research and reiterate my recommendation for acceptance in Nature Communications.

We would like to thank the editor and the three referees for their efforts. Their feedback was very helpful. In the meantime, we have addressed all comments. In particular, we have implemented the following two key changes:

- 1. We now provide a better motivation for our study, and we connect it to a wider range of the literature (as suggested by Reviewer #2). These changes make it more clear why our study should be of interest to the broad readership of Nature Communications.*
- 2. We have run further simulations (as requested by Reviewer #3). For these simulations, we systematically vary the threshold required for a good reputation. We find that this threshold has a negligible effect on our results, which further supports the robustness of our findings (see **Figure S4**).*

In addition, we have made several smaller changes. We believe the suggested changes have improved our manuscript considerably, and we thank the referees for triggering these changes. Please find our detailed response below.

Reviewer #1:

This manuscript considers a theoretical analysis of indirect reciprocity under imperfect information. Compared to studies on complete information or public assessment, there are not many studies that deal with such imperfect information. Therefore, the suitable norms under this system have not yet been fully analyzed and determined. In particular, there are some competing results under binary reputation. To add effective evidence to the argument, this manuscript attempted a systematic approach where individuals assign more nuanced (multiple rather than binary) reputations. Their agent-based simulations and theoretical analysis show that such multiple reputations have strong error-correcting properties. Thus, cooperation is resilient if the population adopts any of four norms of the leading eight, even if the information is private and noisy.

This manuscript is an excellent work that provides substantive results for an imperfectly informed version of indirect reciprocity research. The current version meets publication level in many aspects including literature review, motivational and research questions, methods, results, discussion, and presentation. I recommend its publication as it is with my confidence as an expert in the field.

Reply: We are grateful for this encouraging feedback!

Reviewer #2:

This paper explores indirect reciprocity in the presence of quantitative assessment of reputations, where “quantitative assessment” here means that reputations are coded on an integer scale rather than via a simple binary of “good” or “bad”. The scale is then converted into a reputation via a threshold. The authors find that this type of quantitative assessment can improve the resilience of four of the leading eight social norms to invasion by unconditional defectors or cooperators, even when reputation information is private and noisy. In particular, the use of quantitative assessment helps improve the resilience of norms of indirect reciprocity by providing a mechanism for rapid error correction. This paper contains several interesting results that will be useful to those studying models of indirect reciprocity.

Reply: Thank you for this positive assessment.

Whether these results will be of wide interest, i.e. of interest to the readership of Nature Communications, is not clear. I personally find the technical details of the model very interesting. However the main impact of the results presented here is to show that imperfect information isn't a problem if reputations are quantitative. And so a model of indirect reciprocity fixes a problem with a previous model of indirect reciprocity. The main justification for publishing these models in high impact journals seems to be that they contradict the findings of previous versions of the same model, also published in high impact journals. The snake eats its own tail. Why should someone not working on this specific type of model care about these findings? The authors make some efforts in this direction in the discussion (e.g. line 338-348) but it is not explored in depth, and in any case, the next iteration of the indirect reciprocity model may well show something different. I see enormous value this type of work, but fundamentally this is a technical paper about an extension of a previously published mathematical model.

Reply: Thank you for raising this issue. It seems to us the reviewer's comment touches upon two questions simultaneously.

The first question is: Why should one be interested in models of indirect reciprocity? Traditionally, this field has emerged as a potential mechanism for the evolution of cooperation. However, to us, one of the major appeals of this field is that it allows researchers to address problems in moral philosophy with the tools of evolutionary game theory. This allows researchers to make an informed argument to address questions like: "What should we think of a good person who defects against a bad person?" The literature of indirect reciprocity would argue that such a defection should be considered as "justified". If a social norm is to sustain cooperation, this social norm ought to have the property that people with a bad reputation should be treated differently. Apart from this theoretical contribution, models of indirect reciprocity also have practical implications. For example, especially for online platforms, reputation systems can be explicitly designed. A natural question then becomes how fine-grained reputations should be. Our model helps to identify possible advantages and disadvantages of more nuanced reputations. Overall, we would thus argue that indirect reciprocity has many natural connections to different fields, including evolutionary biology, ethics, social psychology, and mechanism design. After all, questions such as how to define goodness, how to form opinions about others, or how to recover efficiently from loss of reputation are all key questions in a wide range of fields.

The second question is: How reliable is the framework of indirect reciprocity to make robust predictions?

Here, we agree that the literature on indirect reciprocity has not always followed a straight trajectory. One example is given by the very first paper that established the field, Nowak & Sigmund's "Evolution of indirect reciprocity by image scoring" (Nature 1998). The particular Image Scoring strategy introduced in that study was later shown to be unstable (Leimar & Hammerstein 2001). However, in our opinion, such an apparent "contradiction" does not devalue the field; it only shows that original solutions may need to be revised and further improved over time. In this way, past research has shown that the stability of moral systems critically depends on how reputations are disseminated in a population, and how robust social norms are to errors. We contribute to this research by exploring social norms in which individual reputation can go beyond the previously considered categories of "good" and "bad".

Having said that, we would like to clarify that it was not our aim to simply make a modification to the leading eight norms, in order to fix a problem with a previous model. Rather, we introduce a model that allows us to incorporate an important feature of many natural reputation systems, namely that reputations come in different degrees. As a positive side effect of this shift from black-and-white thinking to more fine-grained opinions, we find that cooperation happens to become more robust.

Changes: We have modified our Introduction to appeal to a broader audience, and have extended our Discussion section to better reflect the connection of our work to a number of different fields, following the arguments above. We cite a broader range of literature from various domains to make the relevance of both our setting as well as our findings to scientists not working on indirect reciprocity more explicit. At the same time, we now make it clearer in the Abstract, Introduction and Discussion that our results are a natural consequence of modeling a more realistic reputation system, rather than approaching a previous model with the intent of fixing it.

Reviewer #3:

Research context: The paper explores how cooperative behavior can be sustained in the framework of indirect reciprocity in the absence of perfect information. Direct and indirect reciprocity are the two classical models in game theory to study social dilemmas. Indirect reciprocity is based on developing social norms of reputation, dependent on indirect interactions, that can sustain cooperation in social dilemmas.

Previous results: There has been extensive research in the area of indirect reciprocity. A fundamental theoretical result from Ohtsuki-Iwasa characterizes third-order strategies that can sustain cooperation in the ideal scenario when the reputation information is precise and perfectly synchronized information. These strategies are called leading-eight strategies. However, it has been shown in a PNAS paper that in the realistic scenario when there is noisy information or imperfect information, then the leading-eight strategies either do not evolve or are not able to sustain cooperation. Thus for the realistic scenario of indirect reciprocity with imperfect information mechanisms to sustain cooperative behavior is an important problem, which this work addresses.

Summary of results: The main result of the paper shows that while classical leading-eight strategies consider binary reputations, generalizing them to have quantitative reputations help to sustain cooperation in indirect reciprocity with imperfect information. Also surprisingly quantitative reputation does not help all leading eight strategies, but only four of the leading eight strategies.

Methods: The methods of the paper include simulation results for (a) fixation probability (Figure 3), (b) cooperation rates (Figure 4), and (c) the impact of varying reputation levels (Figure 5). The simulation results are complemented with two analytical results: (a) recovery time analysis for errors, and (b) a nice extension of Ohtsuki-Iwasa result for characterization of strategies in presence of noise which identifies four of the leading-eight strategies as useful.

Importance and novelty: I consider these results very important and appealing as they present a significant advancement of obtaining a mechanism to sustain cooperation for the fundamental model of indirect reciprocity with imperfect information. The results are also novel as the mechanism proposed is very natural and established with simulation and analytical results.

The manuscript is clearly written and presents new fundamental results which are comprehensive and novel. Hence I recommend publication in Nature Communications.

Reply: We appreciate the positive assessment!

Suggestion for improvements for revision:

It would be useful if there is a figure to illustrate the role of varying the threshold for cooperation. The authors show the role of varying reputation levels (Figure 5), and it would be useful to know whether varying the threshold level for cooperation also impacts the cooperation rate.

Reply: This is a very good point. In our model, individual reputations are measured on an integer scale. To make their decisions, individuals need to convert these integer reputations into a binary assessment of either “good” or “bad”. They make this conversion by using a threshold. A reputation score above the threshold is deemed as good, whereas a score below the threshold is treated as bad. While we have systematically varied the range of possible integer scores in our original manuscript, we have not explored the impact of different threshold values.

To address this issue, we have run further simulations. For these simulations, we vary the possible threshold values to cover the full range of possible reputation scores $[-R, R]$ (as in our baseline model, we use $R=5$). For the four leading-eight strategies we previously identified as successful, we find that the cooperation rate in equilibrium remains largely unchanged. Deviations only arise when threshold values become extreme (i.e. when the threshold comes close to $-R$ or R). For the other four leading-eight strategies, we observe very little cooperation independent of the precise threshold used.

*Changes: Our simulation results are illustrated in the new **Figure S4**. We provide a brief discussion of these findings in the Results section of the main text.*

Apart from this, I am happy to congratulate the authors to excellent research and reiterate my recommendation for acceptance in Nature Communications.

We are very grateful for this encouraging and constructive feedback!

REVIEWERS' COMMENTS

Reviewer #1 (Remarks to the Author):

I have confirmed that the revised version has considered all comments by the reviewers. I recommend its publication as it is.

Reviewer #3 (Remarks to the Author):

The authors have revised their manuscript comprehensively and with love to detail. I warmly recommend publication in present form.

We would like to thank the editor and the two referees for their efforts.

Reviewer #1:

I have confirmed that the revised version has considered all comments by the reviewers. I recommend its publication as it is.

Reply: We are grateful for this encouraging feedback!

Reviewer #3:

The authors have revised their manuscript comprehensively and with love to detail. I warmly recommend publication in present form.

Reply: We appreciate this positive assessment!